# Rethinking Teacher-Student Curriculum Learning through the Cooperative Mechanics of Experience

**Manfred Diaz**  *diazcabm@mila.quebec*
*Mila, University of Montreal*

**Liam Paull**  *paulll@mila.quebec*
*Mila, University of Montreal*
*Canada CIFAR AI Chair*

**Andrea Tacchetti**  *attachet@google.com*
*Google DeepMind*

**Reviewed on OpenReview:** *https://openreview.net/forum?id=qWh82br6KT*

## Abstract

Teacher-Student Curriculum Learning (TSCL) is a curriculum learning framework that draws inspiration from human cultural transmission and learning. It involves a teacher algorithm shaping the learning process of a learner algorithm by exposing it to controlled experiences. Despite its success, understanding the conditions under which TSCL is effective remains challenging. In this paper, we propose a data-centric perspective to analyze the underlying mechanics of the teacher-student interactions in TSCL. We leverage cooperative game theory to describe how the composition of the set of experiences presented by the teacher to the learner, as well as their order, influences the performance of the curriculum that is found by TSCL approaches. To do so, we demonstrate that for every TSCL problem, an equivalent cooperative game exists, and several key components of the TSCL framework can be reinterpreted using game-theoretic principles. Through experiments covering supervised learning, reinforcement learning, and classical games, we estimate the cooperative values of experiences and use value-proportional curriculum mechanisms to construct curricula, even in cases where TSCL struggles. The framework and experimental setup we present in this work represents a novel foundation for a deeper exploration of TSCL, shedding light on its underlying mechanisms and providing insights into its broader applicability in machine learning.

## 1 Introduction

Controlling the sequence of tasks that a learning algorithm is exposed to through curriculum has been shown to potentially enhance learning efficiency (Elman, 1993; Krueger & Dayan, 2009; Bengio et al., 2009). One widely used curriculum framework, known as Teacher-Student Curriculum Learning (TSCL) (Graves et al., 2017; Matiisen et al., 2020), specifically gives a *teacher algorithm* the ability to control this sequence. While it is commonly understood that presenting tasks with increasing difficulty can improve learning, the underlying dynamics and structure of teacher-student interaction in this context are still relatively unexplored. Very few works have attempted to understand *when*, and *how* TSCL works (Lee et al., 2021; Wu et al., 2020) while most have focused on providing algorithmic improvements to the problem (Portelas et al., 2019a; Turchetta et al., 2020; Liu et al., 2020; Feng et al., 2021). In this paper, we propose a novel *data-centric* perspective (Ng, 2021) to understand and analyze TSCL algorithms.

We begin by formalizing a general notion of *units of experience* to describe the teacher algorithm's control objects (consumed by the learner). Subsequently, our approach draws inspiration from work on feature attribution (Patel et al., 2021), data valuation (Ghorbani & Zou, 2019; Yan & Procaccia, 2021) and explainability (Lundberg & Lee, 2017), and leverages tools from cooperative game theory (Von Neumann & Morgenstern, 1944; Shapley, 1952) to analyze *how* the

compositions of these units impact teacher-student interactions. We show that, for every TSCL problem, there exists an equivalent cooperative game where *units of experience* are players and teacher-student interactions approximate a sequential coalition formation process (Sec. 4). As a result, the learning progression objective (Schmidhuber, 1991; Oudeyer et al., 2007; Graves et al., 2017) and the teacher bandit policy (Gittins, 1979; Matiisen et al., 2020), two essential components of TSCL, have alternative interpretations as an approximation of player (unit) marginal contribution (Weber, 1988) and a fair allocation mechanism, respectively (Sec. 4.2 & 4.3). Furthermore, because the *order matters* in the case of curriculum learning (Krueger & Dayan, 2009; Bengio et al., 2009), traditional cooperative game-theoretic arguments produce unintuitive results (Nowak & Radzik, 1994). Thus, we leverage *generalized cooperative games* and their solution concepts (Nowak & Radzik, 1994; Sanchez & Bergantiños, 1997) to overcome these limitations and formally extend these data-centric game-theoretic formulations to the curriculum learning setting.

To demonstrate the predictive power and range of problems where this game-theoretic and data-centric interpretation of TSCL applies, we build an experimental setting that evaluates the prospect of cooperation among *units of experience* in problems spanning supervised learning (SL), reinforcement learning (RL), and classical games (Sec. 5). These experiments simulate ordered and unordered coalition formation processes and approximate the cooperative games we developed to describe TSCL. For every problem, we estimate units *a priori* value (e.g., Shapley or Nowak & Radzik values) and demonstrate that these *a priori* values, although expensive to compute (Deng & Papadimitriou, 1994), are useful proxies to find curricula. To this end, we design unordered and ordered value-proportional curriculum mechanisms inspired by value-proportional allocations (Bachrach et al., 2020). In most settings, the unordered mechanism fails to find a reasonable curriculum, demonstrating the unsuitability of traditional game-theoretic tools for the TSCL problem. However, the ordered mechanism consistently finds an optimal or near-optimal ordering (i.e., a curriculum) even when TSCL fails (Sec. 5.2). To understand what impacts the ability of TSCL in those settings, we leverage another cooperative game-theoretic tool, namely, *measures of interactions* (Grabisch & Roubens, 1999; Procaccia et al., 2014), and in particular the Value of a Player to other Player (vPoP) (Hausken & Mohr, 2001), to quantify positive, neutral, or negative pairwise interactions among units. We show that in settings with considerable unit interference, as characterized by their negative pairwise interactions, TSCL cannot produce useful curricula.

## 2 Preliminaries

### 2.1 Cooperative Game Theory

**Cooperative Games.** Cooperative games model problems where players interact to maximize collective gain (Roth, 1988). In a (traditional) cooperative game in characteristic function form among a set of players $\mathbf{U}$, denoted by $\mathcal{G} = \langle \mathbf{U}, v \rangle$, the characteristic function $v : 2^{\mathbf{U}} \to \mathbb{R}$ associates to each coalition $\mathbf{C} \in 2^{\mathbf{U}}$, belonging to the powerset $2^{\mathbf{U}}$, a real number that represents the benefits produced by the players in $\mathbf{C}$ acting jointly. In a cooperative game, a solution concept represents a mechanism that produces allocation vectors $\phi \in \mathbb{R}^{|\mathbf{U}|}$ (Shubik, 1981). Particularly, *Shapley's value* (Shapley, 1952) allocates to each player $\mathbf{u} \in \mathbf{U}$ its average marginal contribution $v(\mathbf{C} + \mathbf{u}) - v(\mathbf{C})$ to coalitions $\mathbf{C} \subseteq \mathbf{U}$, where $\mathbf{u} \in \mathbf{U} - \mathbf{C}$

$$\phi(\mathbf{u}) = \sum_{\mathbf{C}:\mathbf{u} \notin \mathbf{C}} \frac{|\mathbf{C}|!(|\mathbf{U}| - |\mathbf{C}| - 1)!}{|\mathbf{U}|!} \left[ v(\mathbf{C} + \mathbf{u}) - v(\mathbf{C}) \right] \tag{1}$$

and uniquely satisfies the axioms of *efficiency*, *null-player*, *symmetry*, and *linearity*, which are generally considered to be properties of a fair allocation mechanism (van den Brink & van der Laan, 1998).

**Generalized Cooperative Games.** When the order in which players join determines coalitional worth, traditional cooperative games and their solution concepts (e.g., Shapley's value) may produce unintuitive allocations (Nowak & Radzik, 1994). In these games, the generalized characteristic function $v : \mathcal{P}(2^{\mathbf{U}}) \to \mathbb{R}$ assigns to every ordered coalition $\mathbf{C} \in \mathcal{P}(2^{\mathbf{U}})$ in the powerset of permutations $\mathcal{P}(2^{\mathbf{U}})$ its worth if members join in the permutation order. Nowak & Radzik (1994) and Sanchez & Bergantiños (1997) extended Shapley's work and proposed solution concepts for these generalized cooperative games. We focus on the former due to its intuitive formulation

$$\phi_{\text{NR}}(\mathbf{u}) = \frac{1}{|\mathbf{U}|!} \sum_{\substack{\mathbf{C} \in \mathcal{P}(2^{\mathbf{U}}) \\ \mathbf{C}:\mathbf{u} \notin \mathbf{C}}} \left[ v(\mathbf{C} : \mathbf{u}) - v(\mathbf{C}) \right] \tag{2}$$

that averages, for all ordered coalitions $\mathbf{C} \in \mathcal{P}(2^{\mathbf{U}})$ where the unit $\mathbf{u} \in \mathbf{U}$ is appended last, its marginal contribution to the newly formed ordered coalition $\mathbf{C} : \mathbf{u}$.

**Measures of Interactions.** In a cooperative game, a measure of interaction (Grabisch & Roubens, 1999; Procaccia et al., 2014) computes players' influences on other players' outcomes. In particular, we leverage the *value of a player to another player* (vPoP) (Hausken & Mohr, 2001). For the games above, vPoP constructs a matrix whose entries $\phi(\mathbf{u}_i, \mathbf{u}_j) \in \mathbb{R}$ measure the influence player $\mathbf{u}_i$ exerts over player $\mathbf{u}_j$. It measures how the Shapley value of a unit changes in the absence of another. More precisely,

$$\phi(\mathbf{u}_i, \mathbf{u}_j) = \sum_{\substack{\mathbf{C} \subseteq \mathbf{U} \\ \mathbf{u}_i, \mathbf{u}_j \in \mathbf{C}}} \frac{(|\mathbf{U}| - |\mathbf{C}|)(|\mathbf{C}| - 1)!}{|\mathbf{U}|!} \left[ \phi(\mathbf{u}_j, \mathbf{C}) - \phi(\mathbf{u}_j, \mathbf{C} - \mathbf{u}_i) \right] \tag{3}$$

where $\phi(\mathbf{u}_j, \mathbf{C})$ is the Shapley value of unit $\mathbf{u}_j$ (Eq. 1) in the cooperative game restricted to players in $\mathbf{C}$. This matrix marginal $\phi(\mathbf{u}_i) = \sum_j \phi(\mathbf{u}_i, \mathbf{u}_j)$ corresponds to each player's Shapley value. We extend vPoP to games in generalized characteristic function form by applying Eq. 3 *mutatis mutandis* using Nowak & Radzik (1994) value to provide an ordered pairwise interaction metric $\phi_{\text{NR}}(\mathbf{u}_i, \mathbf{u}_j)$.

## 2.2 Bandit Algorithms

Multi-armed bandit algorithms provide a solution to problems of decision-making under uncertainty (Gittins, 1979; Lattimore & Szepesvári, 2020) where, at each interaction, a decision must be made about which arm $\mathbf{u} \in \mathbf{U}$ must be pulled. We are particularly interested in action-value-based algorithms that maintain empirical value estimates $q_k(\mathbf{u})$ computed as

$$q_k(\mathbf{u}) \approx \frac{1}{N_k^{\mathbf{u}}} \sum_{i=1}^{k-1} r(\mathbf{u}_i) \mathbb{I}_{\mathbf{u}_i = \mathbf{u}} \tag{4}$$

and that estimate the average reward received by the algorithm in the iterations $N_k^{\mathbf{u}} \leq k$ where the $\mathbf{u}$-arm has been pulled. Bandit algorithms, like the ones Graves et al. (2017) and Matiisen et al. (2020) use in their work, transform the estimated average contributions into arms interactions by deriving from estimated values a Boltzmann policy $\tau_k \in \Delta(\mathbf{U})$ such that the probability of interaction is proportional to the value estimates:

$$\tau_k(\mathbf{u}) \propto \mathcal{B}(q_k(\mathbf{u})) = \frac{e^{\frac{q_k(\mathbf{u})}{T}}}{\sum_{\mathbf{u}'} e^{\frac{q_k(\mathbf{u}')}{T}}} \tag{5}$$

More sophisticated approaches (e.g., the EXP3 (Auer et al., 2003) used in our experiments) account for other factors, like recency, bias, stochasticity, or non-stationarity (Lattimore & Szepesvári, 2020).

# 3 Experience to Control

The TSCL framework commonly operates under the assumption that tasks presented to a learning algorithm can influence its learning dynamics. Modern iterative learning algorithms process tasks in discrete units. For instance, SL and RL algorithms operate over instances and transitions, respectively. But also, collections of these elementary units, such as batches or episodes, datasets or *environments*, or more generally benchmarks or environment suites, describe a hierarchy of aggregations of experience. Henceforth, we utilize the term ***unit of experience*** for referring to any collection of discrete units that a teacher algorithm can use to control the dynamics of the learner algorithm.

**Example 3.1.** For an analysis, we may define a *unit of experience* as the set of instances of class in a SL classification problem. For example, in the *MNIST* dataset (LeCun & Cortes, 2010), there may be *ten units of experience*, namely, classes ZERO, ONE, TWO, . . . , NINE.

The *units of experience* abstraction indistinctly applies to supervised or reinforcement learning problems. On either paradigm, any iterative learning algorithm is a controllable system whose control inputs are units of experience.

---

**Algorithm 1** Generalized Teacher-Student Curriculum Learning.

---

1: **procedure** GENTSCL
2:     **inputs** policy: $\pi_0$, algorithm: $\mathcal{L}$, units: $\mathbf{U}$, metric: $\mathcal{J}$
3:       *teacher*: $\tau_0 \in \Delta(\mathbf{U})$, *targets*: $\bar{\mathbf{U}}$, budget: $K$
4:     **for** $k = 1 \ldots K$ **do**
5:       $\mathbf{u}_k \sim \tau_k(\mathbf{u})$
6:       $\pi_k \sim \mathcal{L}(\pi_{k-1}, \mathbf{u}_k)$
7:       $r_k \leftarrow \mathcal{J}(\pi_k, \bar{\mathbf{U}}) - \mathcal{J}(\pi_{k-1}, \bar{\mathbf{U}})$
8:       $\tau_{k+1} \leftarrow$ UPDATERULE$(\tau_k, \mathbf{u}_k, r_k)$
9:     **end for**
10:     **output:** $\pi_K$
11: **end procedure**

---

**Example 3.2.** There are four control inputs in mini-batch gradient descent (Goodfellow et al., 2016): the mini-batch $\{x_1, \ldots, x_B\}$, the loss function $\ell$, the parameters $\theta$, and the learning rate $\eta$ such that:

$$\theta_{k+1} = \theta_k - \eta \nabla_{\theta_k} \sum_{i=1}^{B} \ell(\theta_k, x_i)$$

A TSCL-style algorithm, as presented in Alg. 1, solves a data-centric control problem. The *learner* algorithm $\mathcal{L}(\pi_{k-1}, \mathbf{u}_k)$ is a *black-box* system (line 6) controlled by a *teacher* algorithm through units drawn with probability $\mathbf{u}_k \sim \tau_k(\mathbf{u})$. The *learner* output, at each iteration $k$, is policy or model $\pi_k$ whose performance is measured by a *metric function* $\mathcal{J}$ that quantifies the model's performance on a set of evaluation units $\bar{\mathbf{U}}$. The teacher aims to maximize the cumulative learning progression reward (line 7). For the *teacher*'s UPDATERULE, we focus on multi-armed bandit learning (see Sec. 2.2). We adopt a data-centric perspective to perform a systematic investigation of its components.

## 4 The Cooperative Mechanics of Experience

The ideal teacher-student interaction mechanics assume that the *learner* monotonically increases its performance on the target task. We conjecture that a prerequisite for this idealistic curriculum learning dynamics (Matiisen et al., 2020) to occur within TSCL-style algorithms is that experience (or data) presented to the *learner* should not interfere with each other. In other words, *units of experience* should interact cooperatively. From a game-theoretic perspective, we explain how these cooperative mechanics may emerge among units by examining the history of teacher-student interactions, the reward function, and the bandit selection policy.

### 4.1 The Mechanics of Coalition Formation

We establish a cooperative game where each unit of experience $\mathbf{u} \in \mathbf{U}$ is a player. Next, we interpret the history of $k \leq K$ teacher-student interactions $\mathbf{H}_k = \{\mathbf{u}_1, \ldots, \mathbf{u}_k\}$ through their empirical frequencies $p_k(\mathbf{u}) \in \Delta_{\mathbf{U}}$ which form unit vectors that lie in the $|\mathbf{U}|$-probability simplex $\Delta(\mathbf{U})$. The effective support (i.e., non-zero probabilities) determines an unordered coalition (i.e., a set) $\mathbf{C}_k \subseteq \mathbf{U}$ (see Faigle (2022), Chapter 8), formed by the units presented to the *learner* up to interaction $k \leq K$. We study this interpretation through a cooperative game in characteristic function form (Sec. 2.1).

**Example 4.1. (Example 3.1 cont'd)** In the *class-as-unit* equivalence on MNIST, an unordered training coalition, e.g., the two-unit coalition $\mathbf{C} = \{\text{ZERO}, \text{NINE}\}$, describes teacher-student interactions limited to instances from those *classes*.

Next, we note that the outcome of a coalition's work is the policy or model $\pi_k$. Thus, estimating the performance of the policy $\pi_k$ through the metric function $\mathcal{J}$ is akin to approximating the characteristic function $v(\mathbf{C}_k)$ (Alg. 1, line 7). Moreover, these approximations are conditioned on an evaluation (or target) unit $\bar{\mathbf{u}} \in \bar{\mathbf{U}}$. We model the *target-task* and *multiple-task* settings (Graves et al., 2017) where *units of experience* should increase *learner* performance on an evaluation unit (e.g., a task, or an environment) or on multiple evaluation units (e.g., a set of tasks or environments).

| TSCL | Cooperative Game | Learning | Example |
|------|-----------------|----------|---------|
| Units | Players | | $\mathbf{u}_1, \mathbf{u}_2, \mathbf{u}_3$ |
| Sequence | Coalition | | $(\mathbf{u}_2, \mathbf{u}_1)$ |
| Policy Value | Coalitional Worth | $\mathcal{J}(\pi_2)$ | $v(\mathbf{u}_2, \mathbf{u}_1)$ |
| Learning Progression | Marginal Contribution | $\mathcal{J}(\pi_2) - \mathcal{J}(\pi_1)$ | $v(\mathbf{u}_2, \mathbf{u}_1) - v(\mathbf{u}_2)$ |
| Unit Value | Player Value | $q_k(\mathbf{u}_1)$ | $\phi(\mathbf{u}_1)$ |
| Interactions | Allocations | $\tau_k(\mathbf{u})$ | |

Table 1: The mechanics of Teacher-Student Curriculum Learning are equivalent to a cooperative game among units of experience.

Consequently, every notion of coalitional worth is conditional on the evaluation units, thus generating a space of cooperative games.

**Definition 4.1.** (**TSCL Cooperative Games**) Let $\mathbf{U}$ denote a set of *units of experience* $\mathbf{u} \in \mathbf{U}$ and $\bar{\mathbf{U}}$ a set of *evaluation units* $\bar{\mathbf{u}} \in \bar{\mathbf{U}}$. Every evaluation coalition $\bar{\mathbf{C}} \in 2^{\bar{\mathbf{U}}}$ induces a parameterized characteristic function $v_{\bar{\mathbf{C}}}(\mathbf{C}_k) \in \mathbb{R}$ whose value measures the worth of a coalition $\mathbf{C}_k$ when the members of $\bar{\mathbf{C}}$ are the *evaluation units*. Therefore, the TSCL-family of algorithms operate over a parameterized space of cooperative games:

$$\mathcal{G} \langle \mathbf{U}, \cdot \rangle = \left\{ \langle \mathbf{U}, v_{\bar{\mathbf{C}}} \rangle \mid \bar{\mathbf{C}} \subseteq \bar{\mathbf{U}} \right\}$$

comprising $2^{|\mathbf{U}|} \times 2^{|\bar{\mathbf{U}}|}$ possible games and where the *target-task* (i.e., $\bar{\mathbf{C}} = \bar{\mathbf{u}}$) and the *multiple-tasks* (i.e., $\bar{\mathbf{C}} = \bar{\mathbf{U}}$) settings are special cases.

**Example 4.2.** If a *learner* algorithm is presented with units from $\mathbf{C} = \{\text{ZERO}, \text{NINE}\}$ on MNIST, the following condition is expected to hold:

$$v_{\bar{\mathbf{C}} = \{\text{ZERO,NINE}\}}(\mathbf{C}) > v_{\bar{\mathbf{C}} = \{\text{ZERO,ONE}\}}(\mathbf{C})$$

### 4.2 Marginal Contributions to Learning

The notions of coalitions and coalitional worth above induce a game-theoretic interpretation of the learning progression reward. At any iteration $k \leq K$, this reward $r(\mathbf{u}_k) \in \mathbb{R}$ (Alg. 1, line 7) measures the improvement in policy performance after the *teacher* presents a unit $\mathbf{u}_k$ to the *learner* algorithm that produces a new policy $\pi_k \sim \mathcal{L}(\pi_{k-1}, \mathbf{u}_k)$. Thus, we can restate this reward in terms of a game in characteristic function form:

$$r(\mathbf{u}_k) = v(\mathbf{C}_k) - v(\mathbf{C}_{k-1}) = v(\mathbf{C}_{k-1} + \mathbf{u}_k) - v(\mathbf{C}_{k-1}) \tag{6}$$

and note its equivalence to computing the marginal contribution (see Sec. 2 and Eq. 1) of aggregating the unit of experience $\mathbf{u}_k = \mathbf{u}$ to the existing coalition $\mathbf{C}_{k-1}$.

### 4.3 A Fair Allocation Mechanism

A principle of fair attribution in cooperative games is that players get assigned values proportional to their expected marginal contribution. We note that under the learning progression objective, a bandit action-value estimate $q_k(\mathbf{u})$ (Eq. 4) approximates every unit's (or arm's) average marginal contribution after $k$ interactions. Moreover, as discussed in Sec. 2.2, multi-arm bandit algorithms may transform action-values through a Boltzmann projection (Eq. 5) that converts the value estimates into units' probabilities of interaction with the *learner* (i.e., the (stochastic) policy $\tau_k(\mathbf{u})$). Consequently, the units that, up to interaction $k \leq K$, have produced more significant increases on *learner* performance and would be allocated larger fractions of the remaining $K - k$ interactions.

Thus, a multi-armed bandit teacher implements a fair allocation mechanism that computes units' values by approximating their average marginal contributions and converts these approximations into the currency-like utility of the TSCL games, namely, interactions with the learner.

Table 1 summarizes the equivalences between TSCL components and those of a cooperative game we have established throughout this section.

## 5 An Experiment on The Prospect of Cooperation

We design an experimental setting to empirically verify the equivalences we draw between TSCL components and cooperative game-theoretic concepts (e.g., do cooperative solution concepts capture some notion of curriculum?) and to highlight the utility of this data-centric approach to understanding TSCL failure modes. To this end, we incrementally build different parts of the equivalence in supervised learning, reinforcement learning, and classical game settings as follows:

1. For any set of *units of experience* given to the *teacher* algorithm, we simulate their interactions and compute each *a priori Shapley* or *Nowak & Radzik* value (Sec. 5.1).

2. We build two value-proportional mechanisms (i.e., pre-computed teacher policies) leveraging the prior units' values to validate whether a curriculum exists and to show that cooperative solution concepts retrieve such a notion.

3. We leverage units' estimated pairwise interactions to understand the value-proportional mechanism success and TSCL failure from a data-centric perspective (i.e., the composition of the set of units).

### 5.1 A Simulation of Cooperation

We simulate two coalition formation processes where *units of experience* (e.g., classes, environments, or opponents) in each coalition fairly share a finite interaction budget $K \in \mathbb{N}$ and connect the value of the resulting learner's policies with coalitional's worth.

#### 5.1.1 Coalitional Mechanics and Worth

**Cooperative (Non-Ordered) Game.** To simulate a traditional cooperative game (Sec. 2), we design a coalition formation process that draws at each interaction $k \leq K$ a unit $\mathbf{u}_k \in \mathbf{C}$ with uniform probability $\tau_\mathbf{C}(\mathbf{u}_k) \propto |\mathbf{C}|^{-1}$, from a coalition $\mathbf{C}$, and present it to a *learner* algorithm. We compute this procedure for every coalition of units $\mathbf{C} \in 2^\mathbf{U}$. The uniform distribution reflects an *a priori* ignorance of units' importance before measuring their effect on the *learner*.

**Generalized (Ordered) Game.** To test whether our formulation captures order for curriculum, we build a coalition formation process that simulates a generalized cooperative game. In this setting, a *unit* $\mathbf{u} \in \mathbf{C}$ is continually presented to the *learner*, for $\lfloor K/|\mathbf{C}| \rfloor$ interactions, in its permutation order on an ordered coalition $\mathbf{C}$. We repeat this procedure for every $\mathbf{C} \in \mathcal{P}(2^\mathbf{U})$. As before, the ordered equipartition of interactions reflects our ignorance about units *a priori* effect.

**Coalitional Worth & Characteristic Function.** For every coalition, we obtain a model $\pi_\mathbf{C}^K$ after $K$ interactions with the units in $\mathbf{C}$ through either the traditional or generalized mechanics described above. Therefore, by the equivalence we established between policy performance and (conditional) coalitional worth (Sec. 4.2), the evaluation of each policy determines the characteristic function $v(\mathbf{C})$. More importantly, because the resulting policy $\pi_\mathbf{C}^K$ is unbiased for the *evaluation units or coalitions* (e.g., the uninformed priors in the mechanics), we estimate, using the same policy $\pi_\mathbf{C}^K$, the coalitions worth for every characteristic function $v_{\bar{\mathbf{C}}}(\mathbf{C})$ parameterized by every evaluation or target coalition $\bar{\mathbf{C}}$.

**Example 5.1. (Example 4.1 cont'd)** Assume a budget of $K = 100$ interactions and a subset (coalition) of classes from MNIST, for instance, units (classes) ZERO and NINE. In the simulation of a traditional game, for a coalition $\mathbf{C} = \{\text{ZERO}, \text{NINE}\}$, we uniformly draw instances from each unit with probability $\tau(\mathbf{u}) = \frac{1}{2}$. For a coalition $\mathbf{C} = [\text{ZERO}, \text{NINE}]$ in a generalized game, instances from unit ZERO are presented for the first $k = 50$ iterations followed by $k = 50$ instances from NINE. The resulting policy $\pi_\mathbf{C}^K$ performance is equivalent to the value of the characteristic function $v(\mathbf{C})$, evaluated at coalition $\mathbf{C}$.

**Marginal Contributions & Solution Concepts.** The players marginal contributions are the central quantity of cooperative solution concepts. What do marginal contributions capture in our experiments? First, in both coalition formation processes, adding a unit $\mathbf{u}$ to a coalition $\mathbf{C}$ while keeping the number of interactions $K$ constant reduces the learner algorithm's interactions with the existing units. For instance, in the traditional cooperative game simulation, adding a unit $\mathbf{u}'$ reduces the probability of drawing any unit already in $\mathbf{u} \in \mathbf{C}$ from $p_\mathbf{C}(\mathbf{u}) = |\mathbf{C}|^{-1}$ to $p_{\mathbf{C}+\mathbf{u}'}(\mathbf{u}) = (|\mathbf{C}| + 1)^{-1}$, while in the the generalized game reduces units' interactions from $\lfloor K/|\mathbf{C}| \rfloor$ to $\lfloor K/(|\mathbf{C}| + 1) \rfloor$. Therefore,

in either simulation, a unit $\mathbf{u}$ marginal contribution $v(\mathbf{C} + \mathbf{u}) - v(\mathbf{C})$ measures the change in performance produced by increasing learner's interactions with $\mathbf{u}$ while reducing those with the existing units in $\mathbf{C}$.

**Example 5.2. (Example 5.1 cont'd)** In a traditional cooperative game simulation on MNIST, a marginal contribution such as $v(\{\text{ZERO}, \text{NINE}\}) - v(\{\text{ZERO}\})$ measures the change in *learner* performance produced by exchanging approximately $K/2$ interactions with unit ZERO for interactions with NINE. However, in the generalized game simulation, the same expression measures the change in performance produced by exchanging $K$ interactions with unit $\{\text{ZERO}\}$ for $K/2$ with ZERO first (pre-training) followed by $K/2$ with NINE (fine-tuning).

In consequence, both solution concepts leveraging marginal contributions, namely, the *Shapley value* for traditional games (Eq. 1) and the *Nowak & Razik's value* for generalized games (Eq. 2) estimate, in our simulations, a unit's average marginal contribution to learning, and thus capture their helpfulness or cooperativeness for curriculum.

### 5.1.2 A Sanity Check Through Supervised Classification

Our running examples on MNIST inspire the first setting we examine to empirically validate the connections we have established between TSCL and cooperative games. In this experiment, we considered instances aggregated in classes as *units of experience* and benefited from a trained classifier's confusion matrix to extract ground-truth information from unit interactions. For both MNIST and CIFAR10 (Krizhevsky, 2009), we trained a model on the complete dataset (e.g., for 200 epochs), extracted the confusion matrix on validation, and identify the *top-k* most confused pairs of classes. On MNIST, we selected the five classes TWO, THREE, FOUR, FIVE and SEVEN belonging to the top-three most confused pairs (see Appendix A, Fig. 4a), grouped their instances into five *units of experience*, and conduct the approximations described in Sec. 5.1 for a traditional cooperative game (i.e., not considering order). Next, we followed the same approach for CIFAR10 six classes with more significant pairwise confusion errors on validation, namely, CAR, CAT, DEER, DOG, FROG and HORSE (see Appendix A, Fig. 4b).

If the equivalences we drew are correct, one may expect a unit Shapley value to be higher when the evaluation target is the same unit. Moreover, the pairwise interactions between units should approximately reflect the negative interactions we extracted from the confusion matrices of each trained classifier.

**Units' Values.** In effect, for either MNIST or CIFAR10, each unit's Shapley value estimated from the traditional cooperative game simulations correctly matches the ground truth information. In the target-unit setting, where each *unit of experience* is also used as evaluation unit, each *unit of experience* (or class) matching the evaluation unit (or class) has the largest *Shapley value*, as depicted in Fig. 1a (first *five* targets) for MNIST and Fig. 1c (first *six* targets) for CIFAR10. For instance, on MNIST, unit $\mathbf{u} = \text{TWO}$ has the largest *Shapley value* $\phi(\text{TWO}) = 0.995$ when the evaluation unit is $\bar{\mathbf{u}} = \text{TWO}$. We observed a similar effect on CIFAR10. Furthermore, for the *all-units* setting, every unit's *Shapley value* is approximately equal on both MNIST and CIFAR10 (Fig. 1a and 1c, *all* column), matching the intuition that, conditional on an *all* units evaluation, every *unit of experience* should be equally valuable.

**Measures of Interactions.** We also computed the vPoP measure (see Sec. 2.1, Eq. 3) to verify whether its decomposition of *Shapley values* into pairwise interaction values correctly identifies the most confused pairs of classes. For both MNIST and CIFAR10, their respective vPoP matrices, displayed in Fig. 1b and 1d, provide a reasonable approximation to the ground-truth pairwise interactions extracted from the confusion matrices. For instance, on MNIST the *units* TWO and SEVEN have the lowest interaction value $\phi(\text{TWO}, \text{SEVEN}) = \phi(\text{SEVEN}, \text{TWO}) = -0.007$ which corresponds to largest entry $M(2, 7) = 20$ of the confusion matrix (see Appendix A, Fig. 4a). Also, in CIFAR10 the units DOG and CAT have the lowest interaction value $\phi(\text{DOG}, \text{CAT}) = -0.0164$ coinciding with the most confused classes $M(\text{DOG}, \text{CAT}) = 66$ on validation (see Appendix A, Fig. 4b). However, we note that, for instance, on MNIST confusion matrix $M(2, 7) \neq M(7, 2)$ and, similarly, on CIFAR10's $M(\text{DOG}, \text{CAT}) \neq M(\text{CAT}, \text{DOG})$. Nevertheless, we interpret these values as reasonable proxies for units *negative, positive, or neutral* pairwise interactions.

A valuable aspect of our cooperative game-theoretic analysis of TSCL is that the equivalences we introduced are not limited to the supervised classification setting where ground truth information is available. Next, we demonstrate the broad applicability of these ideas through problems in RL and classical games, where finding ground-truth notions of *curriculum* is non-trivial. Details to reproduce these experiments are provided in Appendix A.

Figure 1: We validated the **prospect prior** using the *class-as-a-unit* analogy on MNIST and CIFAR10. In Figures (a) and (c), each column represents units' Shapley values $\phi(\mathbf{u})$ in each cooperative game parameterized by a target-unit $\bar{\mathbf{u}}$ and the target coalition of *all* units. In Figures (b) and (d), we present the vPoP decomposition matrix (Eq. 3) measuring the pairwise interaction values $\phi(\mathbf{u}_i, \mathbf{u}_j)$ among units in the *all-units* target.

### 5.1.3 Reinforcement Learning

We investigate the MINIGRID-ROOMS (Chevalier-Boisvert et al., 2018) set of three environments, namely, TWOROOMS, FOURROOMS, and SIXROOMS for which it is *folk knowledge* that an optimal curriculum exists.[1] We apply the *prospect prior* simulation of a generalized game where we consider each environment a *unit of experience*. As a *learner* algorithm, we used PPO (Schulman et al., 2017) with an interaction budget of $K = 500,000$ steps, and estimated, from the outcome of these simulations, the *Nowak & Radzik* values (Sec. 2, Eq. 2), with every environment and a uniform distribution over *all*, as evaluation targets. In Fig. 2a, we show that the *Nowak & Radzik* values we estimate match *folk knowledge*. First, there is no requirement for environments other than TWOROOMS as the only positive value $\phi_{\bar{\mathbf{u}}}(\text{TWOROOMS}) = 0.423$ correctly measures. Then, for FOURROOMS, the values of $\phi_{\bar{\mathbf{u}}}(\text{TWOROOMS}) = 0.041$ and $\phi_{\bar{\mathbf{u}}}(\text{FOURROOMS}) = 0.107$ indicate that both environments are required. And finally, environments values of $\phi_{\bar{\mathbf{u}}}(\text{FOURROOMS}) = 0.03$ and $\phi_{\bar{\mathbf{u}}}(\text{SIXROOMS}) = 0.03$ indicate that both are needed for solving SIXROOMS.

### 5.1.4 Classical Games

We introduce an experimental setting, the *Adversarial Sparse Iterated Prisoner's Dilemma* (A-SIPD), that utilizes the Prisioner's Dilemma (Flood, 1952; Axelrod & Hamilton, 1981) classical two-player game as a base but in a more challenging *sparse* and iterated version where at the end of a finite number of interactions (e.g., 200 steps), a *win-draw-loss* reward is given to the learner if it accrues more cumulative *payoff* than its opponent. Opponents are drawn from a *population* of five well-known strategies: ALWAYSCOOPERATE, ALWAYSDEFECT, WINSTAYLOSESWITCH, TITFORTAT (Axelrod, 1981) and a ZERODETERMINANT strategy (Hilbe et al., 2013). We apply the *prospect prior* simulation of a *generalized game* where we consider each *opponent* a *unit of experience*. As a *learner* algorithm, we used PPO (Schulman et al., 2017) with a budget of $K = 100,000$ interactions.

We estimated the *Nowak & Radzik* values (Sec. 2, Eq. 2), conditioned on each opponent and a uniform mixture over *all*, as evaluation targets. The results we present in Fig. 2c show that playing uniquely against TITFORTAT is sufficient across all evaluation targets, including the most challenging opponents, ALWAYSDEFECT and ZERODETERMINANT. This result contrasts with *folk knowledge* in population-based training (e.g., playing against the population's *Nash strategy* (Nash, 1950)). We defer to Sec. 5.3 a more in-depth discussion of this finding.

## 5.2 Curriculum from A Priori Values

For both the RL and classical games settings, the Nowak & Radzik value correctly *ordered* tasks and opponents, respectively, by their contribution to learning. Nevertheless, we also investigated whether these values' magnitude indicated the proportion of interactions (i.e., a fraction of the $K$ interactions budget) that should be allocated to each unit and whether the combination of order and magnitude retrieved an approximate curriculum.

Therefore, inspired by value-proportional allocations (Bachrach et al., 2020), we developed two mechanisms that turn units' values into interactions with the *learner* by projecting any value vector $\phi(\mathbf{u}) \in \mathbb{R}^{|\mathbf{U}|}$ onto vectors $\tau(\mathbf{u}) \in \Delta_{\mathbf{U}}$ in a $|\mathbf{U}|$-*simplex*. While Bachrach et al. (2020) use a *linear* projection unable to handle negative marginal contributions, we

---

[1]The curriculum order between these three environments in RL is an intuition that, to our knowledge, has not been quantified before.

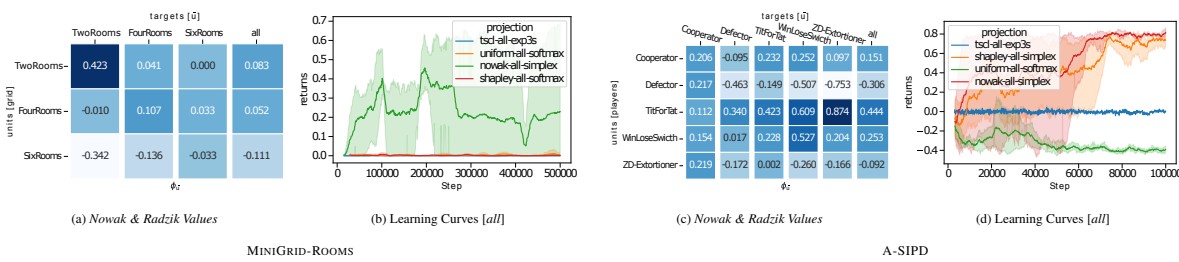

Figure 2: *Nowak & Radzik* values *(a, c)* conditional on each *single-unit* and the *multiple-units* (all) evaluations, and the *learning curves (b,d)* for our mechanisms and TSCL.

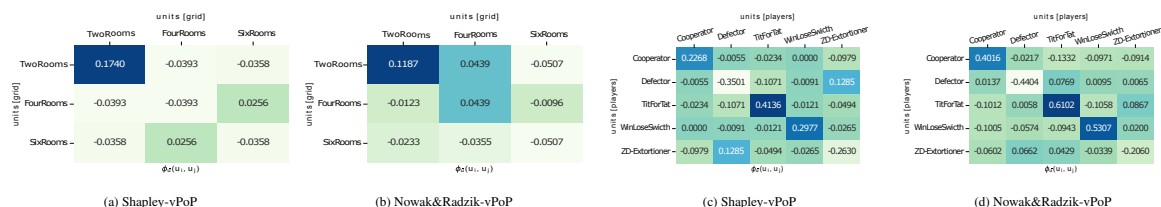

Figure 3: The vPoP decomposition of *Shapley's* and *Nowak & Radzik values*, conditioned on the *all-units* evaluation target, for the MINIGRID-ROOMS *(a, b)* and *A-SIPD (c, d)* problem settings.

investigate the *Boltzmann* or *softmax* projection, commonly used in *multi-arm bandit* algorithms (see Sec. 2.2, Eq. 5), and an *Euclidean* projection (Blondel et al., 2014) that projects to zero any unit with negative value.

When values $\phi(\mathbf{u})$ are *Shapley values*, the projected vectors are used as pre-computed *teacher* policies (i.e., probability distributions), mimicking TSCL's interactions but fixed *a-priori* knowledge of units' value. However, when values $\phi(\mathbf{u})$ are *Nowak & Radzik*, vectors $\tau \in \Delta_{\mathbf{U}}$ are used as *ordered compositional vectors* (Aitchison, 1982) that represent ordered fractions of $K$ interactions. Thus, we construct a pre-computed *teacher* policy that, first orders units by their *Nowak & Radzik values*, projects the ordered values onto $\tau \in \Delta_{\mathbf{U}}^K$, and presents the *i-th ranked unit* $\mathbf{u}_i \in \mathbf{U}$ to the *learner* for the number of interactions indicated by $\tau_i \in \mathbb{N}$. This mechanism preserves the ordered values captured by *Nowak & Radzik*'s solution concept.

Fig. 2 shows that for both MINIGRID-ROOMS and A-SIPD the *teacher* policies obtained from these *value-proportional mechanisms*, in particular the Euclidean projection of *Nowak & Radzik* values (*nowak-all-simplex*), consistently produce learner-induced policies that solve the target tasks, and empirically validate our hypothesis that *a priori* values, in particular the generalized Nowak & Radzik value we introduced, retrieve both the order of and the proportions of interactions that should be allocated to each unit to produce curriculum.

**TSCL Data-Centric Failures.** Moreover, Fig. 2 also highlights the multi-arm bandit approach (EXP3 (Auer et al., 2003; Graves et al., 2017)) to TSCL (i.e., *tscl-all-exp3s*) failure to produce an effective curriculum. We found that in the presence of units with non-cooperative interactions, bandit-driven TSCL fails. Fig. 3 presents the vPoP decomposition of *Shapley* and *Nowak & Radzik* values conditioned on the *all-units* evaluation. In both settings, the interactions measured from the *Shapley*-based decomposition produce lower (negative) values than those obtained with *Nowak & Radzik*. As we show through Sec. 4 and Sec. 5, we take the *Shapley*-based interaction values as fair approximations of TSCL interactions, and thus they provide a data-centric explanation to TSCL failures (see Appendix B). On the other hand, the *Nowak & Radzik*-based values explain the success of *nowak-all-simplex* mechanism, along with the elimination, through the *Euclidean* projection, of negatively-valued units. These pruning strategies are employed effectively by data valuation techniques (Yan & Procaccia, 2021).

## 5.3 Single and Population-based Curriculum

These results offer an alternative approach to what population-based training approaches prescribe as curriculum (Lanctot et al., 2017; Balduzzi et al., 2019; Garnelo et al., 2021; Liu et al., 2022). Generally, *meta-strategy solvers* for population-based training leverage tools from non-cooperative game theory (Von Neumann & Morgenstern, 1944) to find, for

instance, the mixed *Nash equilibrium* (Nash, 1950) of the *empirical meta-game* (Wellman, 2006) played by the population of opponents. In this sense, we may also understand TSCL as a *cooperative meta-strategy solver* that prioritizes among a fixed population of (non-learning) opponents (units of experience) those that improve the learning progression of a single learning player against one or more opponents of the same population.

In the sparse and iterated version of Prisoner's Dilemma that we introduced, the *Defector* strategy remains the (empirical) game *Nash equilibrium*. However, the ordered prospect prior results in Fig. 2c show that when evaluated on the population Nash $\bar{\mathbf{u}} = Defector$, the largest *Nowak & Radzik* value corresponded to the *TitForTat* strategy $\phi_{\bar{\mathbf{u}}}(TitForTat) = 0.34$. Playing against the *TitForTat* strategy remains the optimal solution across all evaluation targets, meaning that *TitForTat* is the best proxy opponent to learn from and reach the Nash equilibrium strategy. We can interpret this result from two perspectives. First, it could indicate that the sparse, iterated, and overtly adversarial version of the game we constructed is a more complicated problem than the original, and the Nash equilibrium, the *Defector* strategy, is a stronger opponent. However, these results may also indicate that a cooperative approach to meta-strategy selection may improve performance in some scenarios. We believe that this connection warrants further investigation.

## 6 Limitations

The simulations we computed in our experiments are computationally expensive. It is well-established that cooperative solution concepts are NP-hard (Deng & Papadimitriou, 1994; Elkind et al., 2009). However, better approximations are possible (Yan & Procaccia, 2021; Mitchell et al., 2022) although we do not explore them in this work. Consequently, we do not consider the *prospect prior* experiments and the *value-proportional curriculum mechanism* as algorithmic innovations to replace TSCL. They represent a *data-centric* approach to study the limits imposed on TSCL-style algorithms by the (non)cooperative mechanics among units of experience. However, we acknowledge that the mechanics of units' interactions also affect other aspects of TSCL. These aspects may include, for instance, the *teacher*'s credit-assignment problem (Gittins, 1979) or neural networks learning and forgetting dynamics (e.g., Lee et al. (2021)). We control for these factors by keeping them constant in our experiments (see Appendix A) but do not undertake their analysis here. Our work is a starting point for more thorough explorations of TSCL and curriculum learning, their underlying mechanisms and broader applicability in machine learning.

## 7 Related Work

**Curriculum Learning.** Since the seminal works of Elman (1993); Krueger & Dayan (2009) and Bengio et al. (2009), a large body of literature has been produced on curricula for machine learning algorithms. Excellent surveys presented by Narvekar et al. (2022); Wang et al. (2022), and Soviany et al. (2022) offer a comprehensive state of recent advances in the field. Our work closely inspects the TSCL framework concurrently introduced by Graves et al. (2017) and Matiisen et al. (2020). While follow-up works have generally focused on either algorithmic innovations (Weinshall et al., 2018; Portelas et al., 2020; Feng et al., 2021; Liu et al., 2020; Portelas et al., 2019b; Racaniere et al., 2020; Florensa et al., 2017; Campero et al., 2021; Du et al., 2022; Florensa et al., 2018) or evaluation benchmarks (Chevalier-Boisvert et al., 2019; Romac et al., 2021), the *data-centric perspective* (Ng, 2021; Karpathy & Abbeel, 2021) we proposed here is less explored. The work by Wu et al. (2020) empirically verifies the same fundamental questions on *why*, *when*, and *how* curriculum learning works. However, while our work is more specifically focused on TSCL, we provide a game-theoretic grounding that could be further extended to analyze Wu et al. (2020) setting. On the other hand, Lee et al. (2021) specifically explore TSCL but with a focus on the deep neural network dynamics leading to *catastrophic forgetting* McCloskey & Cohen (1989). The evaluation mechanisms that we propose in the *prospect prior* simulation and the game-theoretic characterization of experience interference we introduce here provide a more general grounding for the problems studied there.

**Machine Learning & Game Theory.** Our work draws much of its inspiration from the tradition of cross-pollination between machine learning and game theory research (Cesa-Bianchi & Lugosi, 2006). In recent years, game theory research has fuelled work in *optimization* (Daskalakis & Panageas, 2018; Jin et al., 2019), *generative modelling* (Goodfellow et al., 2014; Farnia & Ozdaglar, 2020; Mohebbi Moghaddam et al., 2023), or *robustness* (Madry et al., 2018; Huang et al., 2022a). Also, game-theoretic arguments have been used to revitalize long-standing problems in machine learning. For instance, Gemp et al. (2021) reframed the century-old problem of *Principal Component Analysis* (PCA) (F.R.S., 1901; Hotelling, 1933) as the *Nash* equilibrium (Nash, 1950) of a multiplayer game. Recently, Chang

et al. (2020) reformulated sequential decision-making problems (i.e., RL (Sutton & Barto, 2018)) as local economic transactions among specialized self-interested agents. Moreover, and closer to our data-centric approach, Lundberg & Lee (2017) pioneered work on *feature attribution* and *explainable machine learning models* through cooperative game theory. In particular, they approximate Shapley's value (Shapley, 1952) (Eq. 1) to compute the influence *input features* have on model *predictions*. In a way, our work extends Lundberg & Lee (2017) *input-feature-as-a-player* analogy through the notion of *units of experience* and introduces substantial elements to discuss how the cooperative mechanics of experience affects TSCL at different levels of abstraction (i.e., from features to tasks) and varied learning paradigms.

**Active Learning.** As Graves et al. (2017) and Matiisen et al. (2020) noted in their works, TSCL is an experience-prioritization mechanism holding a solid connection to algorithms that actively select experience. The central problem in *active learning* is to decide which experience to present to a learning algorithm by querying an expert or oracle (Settles, 2010). Areas of research that have also adopted this paradigm of *actively sampling experience* include *experience replay* mechanisms in RL Schaul et al. (2016); Andrychowicz et al. (2017); Fedus et al. (2020) where *transitions* stored in a *replay buffer* (Lin, 1992) are sampled according to some prioritization mechanism. This category of active methods fits perfectly under the TU-game among units of experience we introduced in Sec. 4.

**Multitask Learning.** The cooperative game we designed also provides grounding to the problem of determining *which tasks* should be learned together and *what their optimal order is*, originally from the literature of *multitask learning* (Caruana, 1997). The discussion on *experience interference* in Sec. 4, the *prospect prior* simulation and our computation of *ordered* and *unordered* Shapley values, and the vPoP metric are some novel tools we introduce that could shed new light on this problem. Recent work has explored the different avenues of how *task interference* and *ordering* affect learning outcomes (Standley et al., 2019; Shamsian et al., 2023; Zhang et al., 2021; Fifty et al., 2021; Lin et al., 2019). Our work provides a novel grounding that relates easily to these efforts.

**Continual Learning.** Finally, another area of interest to our work is *continual learning* Parisi et al. (2019); De Lange et al. (2022); Mundt et al. (2023). We note that the *ordered prospect prior* simulates a *continual learning* setting. Units of experience are presented to the learning algorithm in order, and their importance (i.e., value in the cooperative game) is estimated using Nowak & Radzik (1994) extension (Eq. 2) to Shapley's value. This perspective offers a principled approach to rank and order tasks by their importance, a critical aspect of the continual learning setting. We believe that future extensions to the ordered prospect prior experiment may be part of a toolbox to understand continual learning benchmarks (Lomonaco & Maltoni, 2017; Lin et al., 2021; Srinivasan et al., 2022), although we do not undertake that task here.

## 8 Conclusions & Future Work

We reexamined TSCL through the lens of cooperative game theory. By drawing inspiration from work on data valuation, feature attribution and explainability, we provide a novel data-centric perspective that re-frames several of its components through alternative cooperative game-theoretic interpretations. Our experiments confirmed the appropriateness of studying TSCL-style under this framework and highlighted the impact of units' cooperative mechanics on this problem. However, we only began to unveil the potential of allocation mechanisms, solution concepts, and measures of interactions to explain some fundamental aspects of TSCL and hope our work inspires an influx of novel game-theoretic approaches to the problem. Future work would explore more theoretically-grounded analysis of this problem through the connection between convex games (Shapley, 1971) and super(sub)modularity in discrete combinatorial optimization (Dughmi, 2009; Bach, 2011; Krause & Guestrin, 2011) and the extension to continuous set of units through values of non-atomic games (Aumann & Shapley, 1974).

## Acknowledgements

MD would like to thank Eugene Vinitsky for insightful feedback on earlier versions of this manuscript. The NSERC Discovery Grant and the Canada CIFAR AI Chair program supported MD and LP. Computing resources provided by Mila, Québec AI Institute, partly enabled this research.

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

# A  Prospect Prior Experiments Details

We provide for all problems, models or policies architectures, algorithm hyperparameters, and other reproducibility details.[2] All models and architectures are implemented with PYTORCH (Paszke et al., 2019), are configured using HYDRA (Yadan, 2019), and fit on a workstation equipped with a 16 GB *NVIDIA RTX A4000* GPU, 32 GB of RAM, and 32 CPU cores.

## A.1  Supervised Learning

**MNIST.** We trained a model on the MNIST (LeCun & Cortes, 2010) supervised *10-digits* classification task. Specification of the model architecture and hyper-parameters selection are provided in Table 2.

| Hyperparameter | Value |
|---|---|
| *optimizer* | ADAM (Kingma & Ba, 2015) |
| *learning-rate* | $10^{-4}$ |
| *betas* | $(0.9, 0.999)$ |
| *eps* | $10^{-8}$ |
| *batch-size* | 4 |
| *epochs* | 200 |
| *shuffle* | *Yes* |

| Model |
|---|
| CONV2D(32, 3, 1) |
| RELU() |
| CONV2D (64, 3, 1) |
| RELU() |
| MAXPOOL2D (2, 2) |
| DROPOUT(0.25) |
| FLATTEN() |
| LINEAR(9216, 128) |
| RELU() |
| DROPOUT(0.5) |
| LINEAR(128, 10) |

Table 2: Details on the learning algorithm hyperparameters (*left*) and model architecture (*right*) used in the MNIST (LeCun & Cortes, 2010) experiments. Model components and the optimizer are provided by PYTORCH (Paszke et al., 2019). These details remained constant throughout the rest of the experiments with MNIST.

**CIFAR10.** The experiments on CIFAR10 (Krizhevsky et al., 2009) follow the same setting as those on MNIST. We similarly trained a model on the supervised *10-classes* task. Specification of the model architecture and hyperparameter selection are provided in Table 3.

| Hyperparameter | Value |
|---|---|
| *optimizer* | ADAM (Kingma & Ba, 2015) |
| *learning-rate* | $10^{-4}$ |
| *betas* | $(0.9, 0.999)$ |
| *eps* | $10^{-8}$ |
| *batch-size* | 4 |
| *epochs* | 200 |
| *shuffle* | *Yes* |

| Model |
|---|
| CONV2D (3, 6, 5) |
| RELU() |
| MAXPOOL2D (2, 2) |
| CONV2D (6, 16, 5) |
| RELU() |
| MAXPOOL2D (2, 2) |
| FLATTEN() |
| LINEAR(400, 120) |
| RELU() |
| LINEAR(120, 84) |
| RELU() |
| LINEAR(84, 10) |

Table 3: Details on the learning algorithm hyperparameters (*left*) and model architecture (*right*) used in the CIFAR10 (Krizhevsky et al., 2009) experiments. Model components and the optimizer are provided by PYTORCH (Paszke et al., 2019). These details remained constant throughout the rest of the experiments with CIFAR10.

---

[2] Regardless, we plan to release the complete source code of all our experiments.

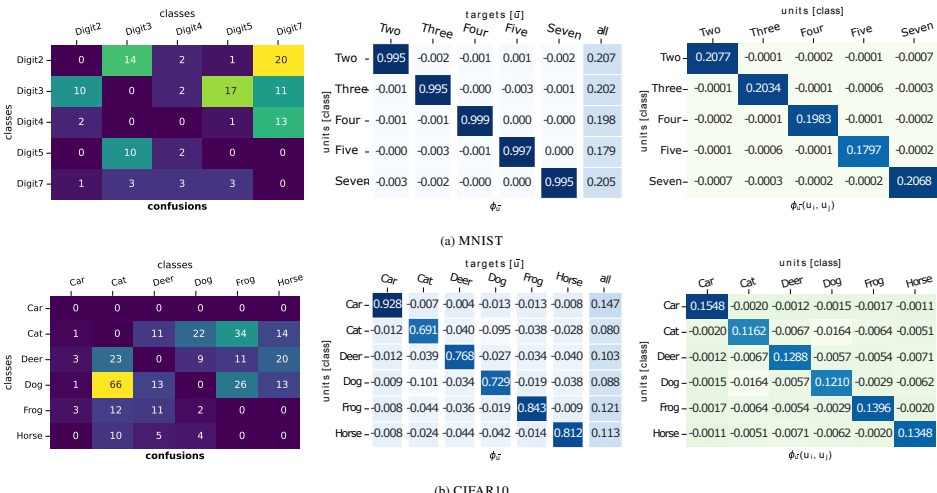

Figure 4: The *class-as-a-unit* analogy applied to MNIST (a) and CIFAR10 (b) served as our ground truth. For each problem, we derived the Shapley's value from the precomputed priors (*left*) [Eq. 1] on each *cooperative game* (Sec. 5). Our results verify that units values on the *target-unit* settings *approximately* ordered the most confused pairs of classes. For instance, digits 2 & 7 in MNIST, or *dog* & cat in CIFAR10. When the target is *all* classes, the vPoP decomposition (*right*) also (Sec. 2.1) identifies *interfering* pairs.

## A.2 Reinforcement Learning

MINIGRID ROOMS. We utilized a sequence of TWOROOMS, FOURROOMS, and SIXROOMS *gridworlds* provided on MINIGRID (Chevalier-Boisvert et al., 2018) as *units of experience*. As the learning algorithm, we trained for $500,000$ steps a PPO Schulman et al. (2017) agent, whose implementation we derived from CLEANRL Huang et al. (2022b). Policy and actor-critic architecture, with shared backbone, as well as other PPO hyperparameters details are presented in Table 4. For the TSCL experiments, we leveraged *Exp3S* (Auer et al., 2003) implementation from Besson (2018) with default hyperparameters $\alpha = 10^{-5}$ and $\gamma = 0.05$, as defined in Graves et al. (2017).

| Hyperparameter | Value |
|---|---|
| *optimizer* | ADAM Kingma & Ba (2015) |
| *learning-rate* | 0.0025 |
| *annealing* | *Yes* |
| *num-steps* | 128 |
| *total-timesteps* | $500,000$ |
| *seeds* | 5 |
| *gamma* | 0.99 |
| *GAE-lambda* | 0.95 |
| *num-minibatches* | 4 |
| *update-epochs* | 4 |
| *advantage-normalization* | *Yes* |
| *clip-value-loss* | *Yes* |
| *clip-coeff* | 0.2 |
| *entropy-coeff* | 0.01 |
| *vf-coeff* | 0.5 |
| *max-grad-norm* | 0.5 |
| *target-kl* | *No* |

| Actor | Critic |
|---|---|
| CONV2D(16, 2, 2) | |
| RELU() | |
| MAXPOOL2D(2, 2) | |
| CONV2D(16, 32, 2, 2) | |
| RELU() | |
| CONV2D(16, 64, 2, 2) | |
| RELU() | |
| LINEAR(64, 64) | LINEAR(64, 64) |
| TANH() | TANH() |
| LINEAR(64, 7) | LINEAR(64, 1) |

Table 4: Details on the PPO hyperparameters (*left*) and *actor-critic* architecture (*right*) used in the MINIGRIDROOMS (Chevalier-Boisvert et al., 2018) experiments. Policy and critic components, and the optimizer, are provided by PYTORCH (Paszke et al., 2019). Implementation and default hyperparameters are derived from CLEANRL (Huang et al., 2022b). These details remained constant throughout the rest of the experiments with MINIGRIDROOMS.

### A.3 Populations & Games.

**Adversarial SIPD.** In our more challenging sparse and iterated version of Prisoner's Dilemma, at the end of $200$ interactions, inspired by Axelrod's competition (Axelrod, 1981). A *win-draw-loss* reward $r = \{-1, 0, 1\}$ is given to a learning player if it beats a fixed opponent. Opponents are drawn from a population of five well-known strategies: *always cooperate*, *always defect*, *win-stay-lose-switch*, *tit-for-tat*, and a *zero-determinant strategy* (Axelrod, 1981; Hilbe et al., 2013; Knight et al., 2021). We trained for 500 episodes (or $100,000$ steps) a PPO (Schulman et al., 2017) agent adapted from CLEANRL Huang et al. (2022b) default implementation. Policy and actor-critic architecture, **without** shared backbone, as well as other PPO hyperparameters details are presented in Table 5. For the TSCL experiments, we leveraged *Exp3S* (Auer et al., 2003) implementation from Besson (2018) with default hyperparameters $\alpha = 10^{-5}$ and $\gamma = 0.05$, as defined in Graves et al. (2017).

| Hyperparameter | Value |
|---|---|
| *optimizer* | ADAM (Kingma & Ba, 2015) |
| *learning-rate* | 0.0025 |
| *annealing* | *Yes* |
| *num-steps* | 128 |
| *timesteps* | $100,000$ |
| *seeds* | 5 |
| *gamma* | 0.99 |
| *GAE-lambda* | 0.95 |
| *minibatches* | 4 |
| *epochs* | 4 |
| *advantage-norm* | *Yes* |
| *clip-value-loss* | *Yes* |
| *clip-coeff* | 0.2 |
| *entropy-coeff* | 0.01 |
| *vf-coeff* | 0.5 |
| *max-grad-norm* | 0.5 |
| *target-kl* | *No* |

| Actor | Critic |
|---|---|
| LINEAR$(2, 64)$ | LINEAR$(2, 64)$ |
| ORTHOINIT$()$ | ORTHOINIT$()$ |
| TANH$()$ | TANH$()$ |
| LINEAR$(64, 64)$ | LINEAR$(64, 64)$ |
| ORTHOINIT$()$ | ORTHOINIT$()$ |
| TANH$()$ | TANH$()$ |
| LINEAR$(64, 2)$ | LINEAR$(64, 1)$ |

Table 5: Details on the PPO hyperparameters (*left*) and *actor-critic* architecture (*right*) used in the ADVERARIAL-SIPD experiments. Policy and critic components, and the optimizer, are provided by PYTORCH (Paszke et al., 2019). Implementation and default hyperparameters are derived from CLEANRL (Huang et al., 2022b). These details remained constant throughout the rest of the experiments.

# B Extended Experiments Results

## B.1 Value-Proportional Curriculum

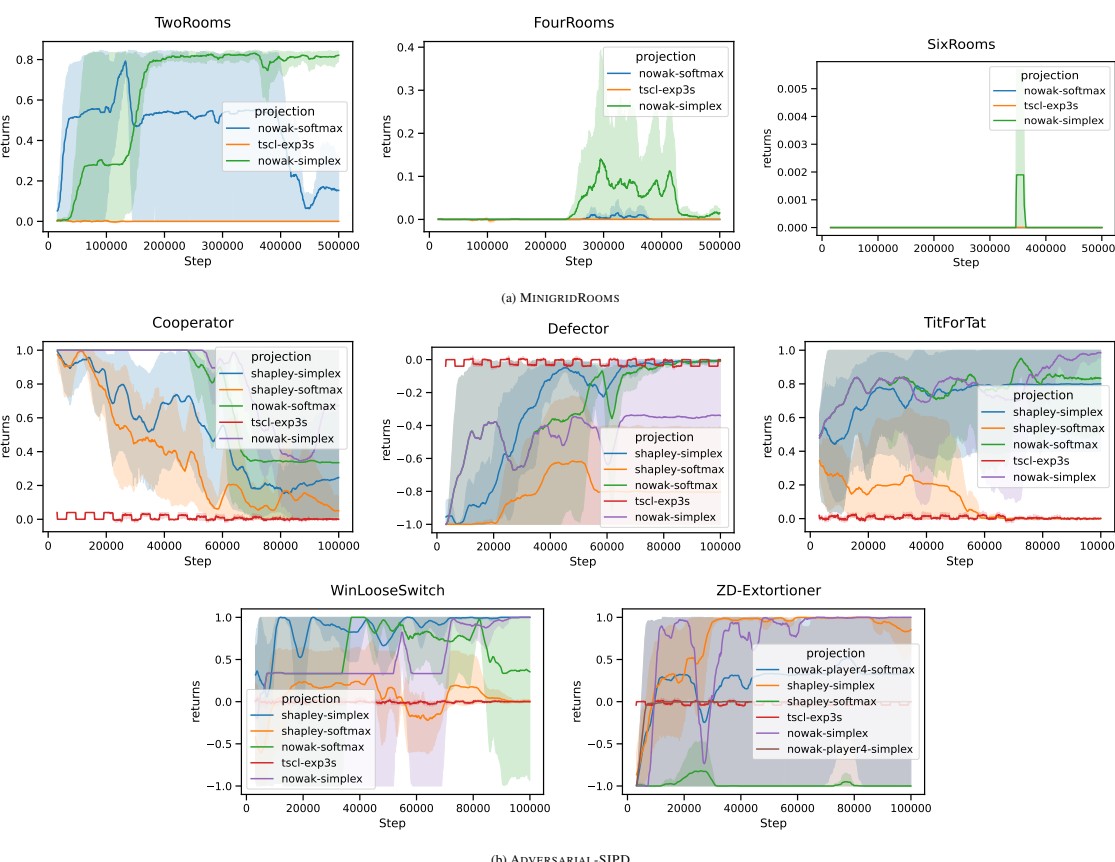

Figure 5: We also investigated the *prior-proportional curriculum* in the *target-unit* setting. For each target unit, we allocate to each training unit interactions proportional to their pre-computed values for each target. For the ADVERSARIAL-SIPD and MINIGRID-ROOMS controlled their learning dynamics by presenting the units according to *unordered* and *ordered* mechanisms in Sec. 5. On each task, the *value-proportional curriculum* derived from the *prospect priot* outperforms TSCL *(tscl-\*-exp3s)*. We further investigate the reason for TSCL failures on this scenario.

## B.2 TSCL Failures

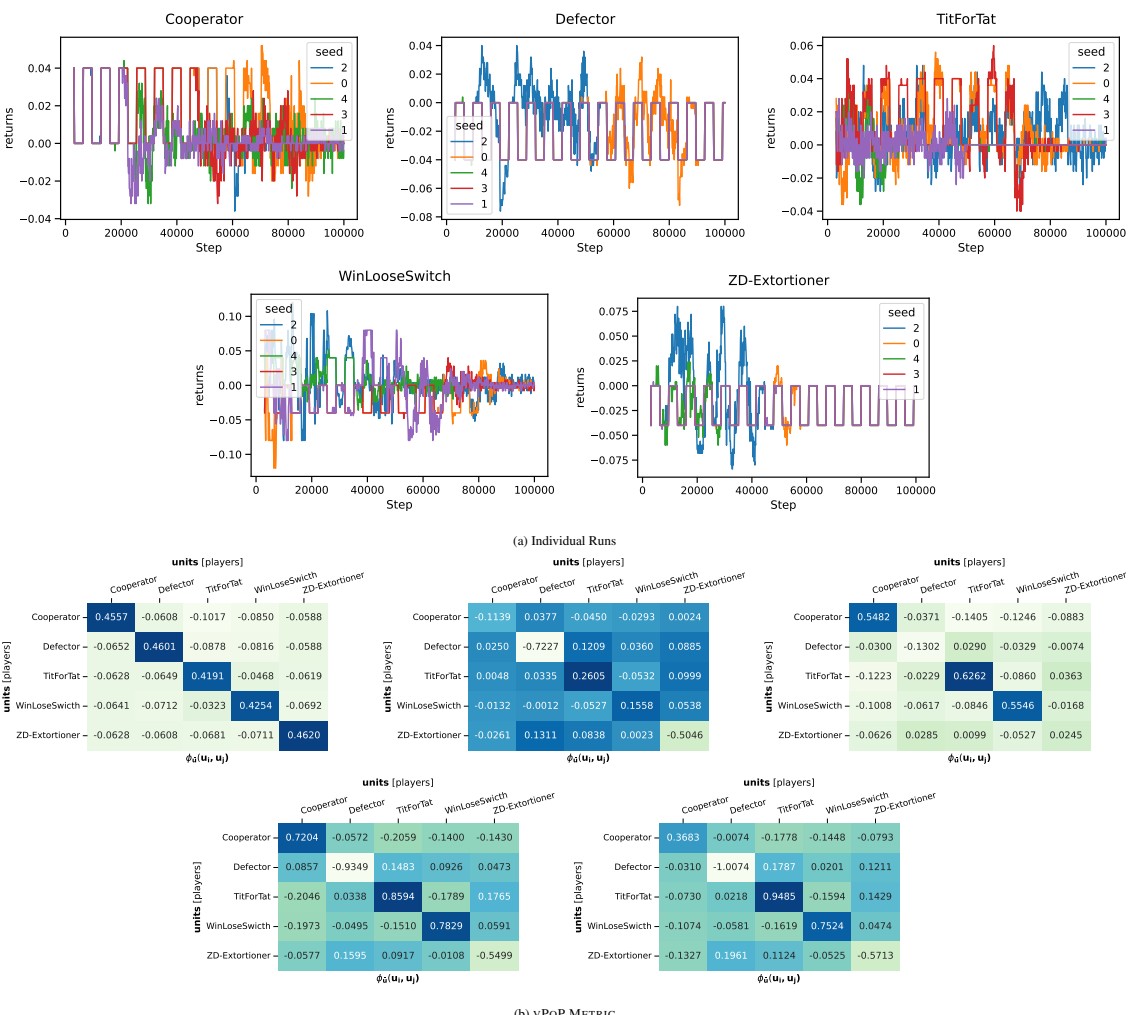

Figure 6: To understand the failure modes of TSCL on ADVERSARIAL-SIPD, we represented the individual runs (i.e., each of the five seeds) on every target unit. TSCL *(tscl-\*-exp3s)* (top row) is extremely brittle, unstable, and generally not robust to units interference. We surmise that these failures are related to the *exploration-exploitation* dilemma. Exploratory steps presenting a negatively-valued unit are hard to overcome (forgetting dynamics). This issue requires further investigation, and we defer it to future work.

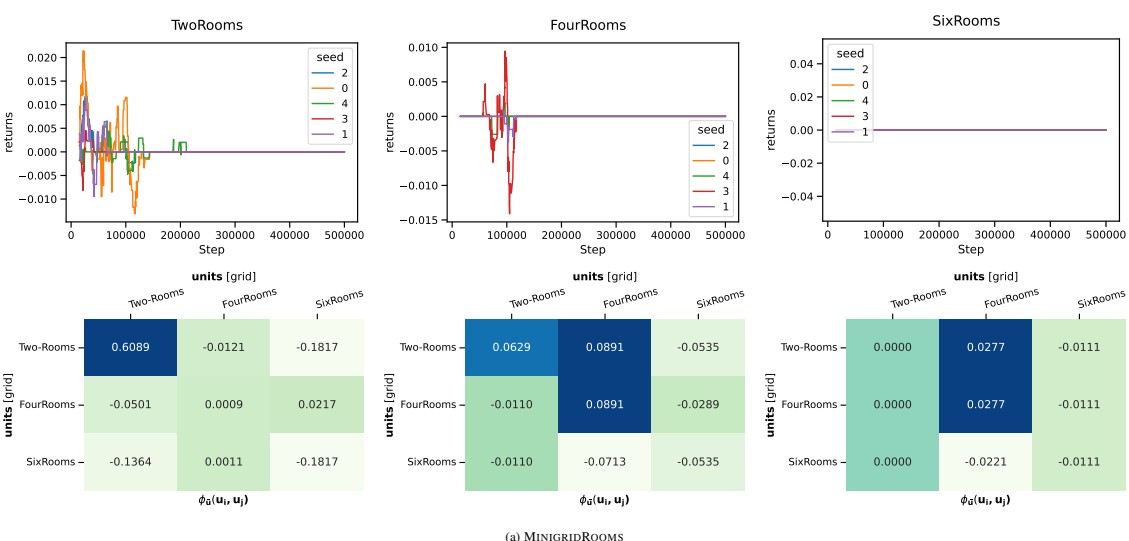

(a) MINIGRIDROOMS

Figure 7: We found that TSCL presents a similar problem in MINIGRIDROOMS. When actions (units) need to be almost deterministically drawn for several steps, and other actions (units) have negative interference with the target, TSCL is unable to find a stable and robust solution to the p roblem.

