# OpenReview forum: "Rethinking Teacher-Student Curriculum Learning through the Cooperative Mechanics of Experience"
_TMLR — Accepted by TMLR_

### Review · Reviewer_UkMA · 2024-06-12

**Summary Of Contributions:**

To understand the conditions under which teacher-student curriculum learning (TSCL) is effective, this paper proposes a data-centric perspective to analyze the underlying mechanics of the teacher-student interactions in TSCL. Authors leverage cooperative game theory to describe how the composition of the set of experiences and orders presented by the teacher to the learner influences the performance of the curriculum. Authors further demonstrate that for every TSCL problem, there exists an equivalent cooperative game. Extensive experiments demonstrates the cooperative values of experiences and use value-proportional curriculum mechanisms to construct curricula.

**Audience:**

Yes

**Broader Impact Concerns:**

The reviewer does not see any major ethical or moral concerns regarding the method or dataset used in the paper.

**Claims And Evidence:**

Yes

**Requested Changes:**

See weaknesses above, especially empirically or theoretically justifying the proposed concepts. More details and results on advanced tasks are needed.

**Strengths And Weaknesses:**

**Strengths**

1. The authors provide helpful illustrations in the figures that support understanding the concepts.

2. This paper extensively reexamines TSCL through the lens of cooperative game theory. The experiments covering supervised learning, reinforcement learning, and classical games provide insights how TSCL can be applied in machine learning.

**Weaknesses**

1. In general, the conducted experiments are restricted to toy datasets (e.g., MINIST and CIFAR10 in the supervised classification setting). The backbones are limited to 2-3 layers convolutional neural networks. It is not clear if TSCL still remains effective for complicated tasks.

2. For each problem, the proposed framework relies on a priori value to find a curricula (e.g., the model's confusion matrix for MINIST). While such priori value is effective, the computation cost can be very expensive with large-scale training samples.

3. Based on the above observations, the TSCL seems to be a post-training learning algorithm for well trained models. It may contain issues such as forgetting dynamics. I am wondering whether authors have any ideas to address such issues.

---

> ### Author Response · Authors · 2024-07-25
>
> We thank the reviewer for the thorough feedback!
>
> __Experimental Scale__. We agree with the reviewer that our selected tasks are relatively simple. As we explain in the General Rebuttal, we could leverage existing approximation whenever scale is a concern to alleviate the computational costs. In our work, we prioritized the accuracy of our estimates by limiting the number of units considered. There is also a practical concern that, as we take up more and more complicated tasks, the existence of a curriculum is more challenging to verify. Moreover, as our experiments highlight, even in those simple tasks, multi-armed bandit approaches to TSCL fail to uncover the existent curriculum. It is not unreasonable to assume that these results hold for more complicated scenarios where the interactions among tasks are more complex.
>
> __TSCL & Post-Training__. There is a distinction between our work and TSCL. First, TSCL is an online approach (e.g., not a post-training approach) to the problem of curriculum learning. The framework we introduced is an a priori analysis tool for the TSCL framework and its issues. Moreover, in Section 5.1, even if the sanity check experiments rely on a classifier’s confusion matrix to empirically validate the defined equivalences,  our method does not require access to such privileged information, as demonstrated in the RL and classical games experiments where such information was unavailable.
>
> __Forgetting Dynamics__. As mentioned in the _Limitations_ section (Section 6), while we control for the effects of forgetting dynamics, we agree with the reviewer that they are a problem for curriculum learning in general and TSCL in particular. We could foresee a few fundamental ideas being readily applicable to this issue in the context of TSCL. For instance, introducing intentional forgetting dynamics into the leaner algorithm [1] could be essential for TSCL to overcome the negative interactions among units that impact early learner models and the credit assignment problems in the teacher algorithm.
>
> [1] Nikishin, E., Schwarzer, M., D’Oro, P., Bacon, P.L. and Courville, A., (2022). The primacy bias in deep reinforcement learning. In International Conference on Machine Learning (pp. 16828-16847). PMLR.

---

### Review · Reviewer_wosX · 2024-07-04

**Summary Of Contributions:**

The paper studies the problem of Teacher-Student Curriculum Learning (TSCL) using tools and ideas from cooperative game theory, which have been used in the past for feature attribution and explainability. In particular, the paper provides a game-theoretic interpretation to many important concepts of TSCL using the notion of units of experience, which are controller by the teacher algorithm; the cooperation between the units of experience is precisely what drives the connection with cooperative game theory. The cooperation between the units of experience is evaluated experimentally in different problems, ranging from supervised learning, reinforcement learning, and game theory.

**Audience:**

Yes

**Broader Impact Concerns:**

No concerns regarding ethical implications of the work.

**Claims And Evidence:**

Yes

**Requested Changes:**

Some minor comments for the authors, which do not have a bearing in the evaluation of the paper:

- Last paragraph in Page 7, there is a typo: "We we"
- Footnotes should come after punctuation marks
- In the concluding section, the authors discuss about the possibility of using super(sub)modularity in discrete combinatorial optimization. I am not able to follow what the authors mean here. Perhaps they can elaborate further.
- How do the authors bypass the fact that many of the solution concepts used in the experiments are NP-hard to compute? Is some approximation used, or do they somehow solve the problem to optimality?

**Strengths And Weaknesses:**

Overall, the paper makes an interesting and natural connection between two different lines of work: Teacher-Student Curriculum Learning (TSCL) and cooperative game theory. To my knowledge, this connection is new, and certainly brings a number of interesting insights to TSCL. It is notable that the results in the paper are not just based on standard ideas from cooperative game theory, such as the Shapley value, but they also use more advanced ideas. For example, the paper of Nowak and Radzik which addresses the problem of order in the coalition formation, not present in the traditional formulation of Shapley; as the authors demonstrate experimentally, that refinement turns out to be particularly crucial in the context of TSCL, in which the order (in terms of units of experience controller by the teacher algorithm) is expected to be make a difference. Another interesting concept leveraged is the measure of interaction, which is used to quantify the pairwise interaction between the units. I found that measure particularly informative in the experiments. So, the paper makes a concrete contribution and provides a number of new insights to an important problem. I also found that the experiments were very interesting. covering a broad range of applications. The paper is also very well-written and organized, and the related work section is quite thorough, discussing many related lines of work.

---

> ### Author Response · Authors · 2024-07-25
>
> We thank the reviewer for the thorough feedback and positive evaluation of our work!
>
> __Computational Hardness__. As mentioned in the General Rebuttal, we know multiple approximations and other relaxations to the problem. Still, they are part of follow-up work that should focus on bringing algorithmic contributions to the issues we uncovered. In our work, to eliminate the stochasticity of the approximations, we solve the problem to optimality by limiting the number of units considered.
>
> __Connections to Discrete Combinatorial Optimization__.  As we briefly mentioned in the Conclusions section (Section 8), there is a strong connection between convex cooperative games and equivalent concepts in discrete combinatorial optimization (e.g., super(sub)modularity, super(sub)additivity [2,3]). For instance, the existence of the Shapley value belongs to the Core (another solution concept) of convex games [1] (convexity being a stronger form of superadditivity). We would like to explore multiple future, more theoretical avenues through these connections.
>
> [1] Shapley, Lloyd S. “Cores of Convex Games.” International Journal of Game Theory, 1971.
>
> [2] Bach, Francis. “Learning with Submodular Functions: A Convex Optimization Perspective.”
>
> [3] Dughmi, Shaddin. “Submodular Functions: Extensions, Distributions, and Algorithms. A Survey.” arXiv [cs.DS], 1 Dec. 2009

---

### Review · Reviewer_TvC9 · 2024-07-12

**Summary Of Contributions:**

This paper describes an approach for using game theory to study Teacher-Student Curriculum Learning (TSCL) and leveraging tools from game theory, as well as incorporating game-theoretic considerations, to understand various problems in machine learning, ranging from supervised learning to reinforcement learning. A number of empirical computational experiments with these domains are conducted in applying the game-theoretic approach.

**Audience:**

Yes

**Broader Impact Concerns:**

There do not appear to be any broader impact concerns that would need to be addressed.

**Claims And Evidence:**

No

**Requested Changes:**

The empirical evaluation is difficult to follow. As described in the Strengths and Weaknesses section, it is not clear what the hypothesis that is being tested in this paper is, and how it could be proven or disproven. For example, the Reinforcement Learning section refers to folk knowledge about the existence of an optimal curriculum, and appears to show estimated allocation values that "match folk knowledge". It's not clear what this means concretely and what insights one can draw from these experiments. Improving the clarity of the experimental evaluation is critical for acceptance.

There are a number of typographical errors, for example:
- beginning of section 2.2: "Multi-armed bandit algorithms...uncertainty" is missing a word, perhaps something like "involving"; "about with" should be "about which"
- bottom of page 7: "we We" should be "we"

**Strengths And Weaknesses:**

The key strength of the work is that this appears to be a novel application of game-theoretic techniques to problems in machine learning, demonstrating a way to interpret concepts from machine learning using the language and formalisms of game theory.

The primary weakness is that the framework appears to be too broad, and insufficiently specific, for deriving actionable insights for analyzing the domains presented here. It would be very useful for the authors to formulate specific hypotheses or claims that can then be subsequently proven or disproven empirically. As currently formulated, it is not clear what insights can be drawn from the experiments.

---

> ### Author Response · Authors · 2024-07-25
>
> We thank the reviewer for the thorough feedback on the manuscript and for highlighting the novelty of our approach!
>
> __Unclear Experimental Hypothesis__. We thank the reviewer for highlighting this shortcoming of our work. As we explain in the General Rebuttal, we have taken steps in the updated manuscript to clarify the central hypothesis that our experiments support. The introduction to Section 5 now clearly states the primary purpose of our experiments. We look forward to hearing from the reviewer on whether these changes alleviate the concerns around the experiments conducted and whether their results support the paper's central claims.
>
> __Framework Scope and Insights__. We agree with the reviewer's concern that the central hypothesis and claims of the paper could have been more clearly stated. As we mentioned in the General Rebuttal, we have taken steps to clarify this issue. However, we believe that the framework's broad scope of applicability is one of its main strengths and contributions.  Very few works have attempted to understand when and how TSCL works [1,2], but none have brought a formal perspective to their analysis. In contrast, we propose a novel data-centric perspective grounded in well-studied cooperative game theory concepts. Moreover, the notions of units of experience and their formulation as players in a cooperative game expand our analysis and TSCL as a curriculum learning framework to different learning paradigms, as we show in our experiments through tasks in supervised learning, reinforcement learning and classical games.
>
> __Other Concerns__. We apologize for the confusion around the “folk knowledge” idea. In the updated manuscript, we clarify that we consider the curriculum order between these three environments in RL to be folk knowledge (i.e., an intuition ) that, according to our understanding, was not quantified before.
>
> [1] Lee, S., Goldt, S., & Saxe, A. M. (2021). Continual Learning in the Teacher-Student Setup: Impact of Task Similarity. International Conference on Machine Learning.
>
> [2] Wu, X., Dyer, E., & Neyshabur, B. (2020). When Do Curricula Work? International Conference on Learning Representations.

---

### Author Response · Authors · 2024-07-25
**General Rebuttal**

We sincerely appreciate the valuable feedback provided by all reviewers. The insights have contributed to enhancing the quality of our work, and we are grateful for their contributions.  To address this feedback, we uploaded a new version of the manuscript reflecting the input provided by all reviewers. Text highlighted in blue indicates fragments we have introduced or modified.


__General Clarification__. We agree with the reviewers that the paper's main hypothesis and central claims could have been more clearly explained. Our work does not contribute a new algorithm to improve curriculum learning or the TSCL. Instead, we introduce an analysis framework for TSCL based on cooperative game theory. The main contribution of our work is a foundational connection between cooperative game theory and TSCL. The equivalences we draw allow us to understand better the data-centric conditions necessary for TSCL algorithms to be effective and provide us with tools to understand their failure modes. We hope it serves as a starting point to understand the problem space better and develop novel algorithmic contributions that address the issues we uncovered. We corrected the language in the experiments section (Section 5) that may have contributed to interpreting our work as an algorithmic innovation in this space (e.g., outperform, compare).


__Experimental Design__. We apologize for the lack of clarity surrounding the experimental design, its goals, and motivation. In the updated manuscript, we clarify that the goals and motivation of our experimental design are:

1. To empirically verify the equivalences we draw between TSCL components and cooperative game-theoretic concepts (e.g., do cooperative solution concepts capture some notion of curriculum?).
2. To highlight the utility of this data-centric approach in understanding TSCL failure modes.


__Computational Cost__. The _Limitations_ section (Section 6) states that cooperative solution concepts are computationally hard problems with well-known complexity classes [1]. Recent advances [2, 3] give us hope of obtaining better approximations of the cooperative game we designed for large-scale problems. Moreover, we also need to highlight that the central hypothesis of curriculum learning (e.g., there is ordering among units) has a combinatorial structure. To verify whether ordering exists (or to prove it does not), one must enumerate all possible orderings and compute their effects on the learner algorithm. Teacher learning algorithms (e.g., the multiarm bandit) help amortize this search, but when they fail, as we demonstrate they do, we are left with no recourse to understand their failure modes. Our work addresses these limitations through cooperative game theory.


[1] Deng, Xiaotie, and Christos H. Papadimitriou. “On the Complexity of Cooperative Solution Concepts.” Mathematics of Operations Research, vol. 19, no. 2, 1994, pp. 257–66

[2] Yan, Tom, and Ariel D. Procaccia. “If You Like Shapley Then You’ll Love the Core.” Proceedings of the... AAAI Conference on Artificial Intelligence. AAAI Conference on Artificial Intelligence, vol. 35, no. 6, May 2021, pp. 5751–59

[3] Mitchell, Rory, et al. “Sampling Permutations for Shapley Value Estimation.” Journal of Machine Learning Research: JMLR, vol. 23, no. 43, 2022, pp. 1–46

---

### Decision · Action_Editor_nXHq · 2024-08-17

**Recommendation:** Accept as is

**Comment:**

The reviewers all agree that the paper has met the criteria for acceptance and state that their questions and concerns regarding the initial submission have been addressed in the revised manuscript.

**Audience:**

The reviewers agree that the connection between Teacher-Student Curriculum Learning and cooperative game theory is novel, interesting, and potentially insightful.

**Claims And Evidence:**

The reviewers agree that the theoretical claims are well-supported and clearly stated. After revisions, the reviewers agree that the experimental claims are clear and sufficiently supported by experimental evidence, though scope of the experiments is somewhat limited by the contrived nature of the problems tested.